# Crystal-Plane and Shape Influences of Nanoscale CeO_2_ on the Activity of Ni/CeO_2_ Catalysts for Maleic Anhydride Hydrogenation

**DOI:** 10.3390/nano12050762

**Published:** 2022-02-24

**Authors:** Shaobo Liu, Xin Liao, Qiuming Zhang, Yin Zhang, Hao Wang, Yongxiang Zhao

**Affiliations:** Engineering Research Center of Ministry of Education for Fine Chemicals, School of Chemistry and Chemical Engineering, Shanxi University, Taiyuan 030006, China; lovog@163.com (S.L.); lx2006294034@126.com (X.L.); matthwwe@163.com (Q.Z.)

**Keywords:** maleic anhydride, crystal-plane, oxygen vacancies, hydrogenation, Ni/CeO_2_

## Abstract

Through use of the hydrothermal technique, various shaped CeO_2_ supports, such as nanocubes (CeO_2_-C), nanorods (CeO_2_-R), and nanoparticles (CeO_2_-P), were synthesized and employed for supporting Ni species as catalysts for a maleic anhydride hydrogenation (MAH) reaction. The achievements of this characterization illustrate that Ni atoms are capable of being incorporated into crystal lattices and can occupy the vacant sites on the CeO_2_ surface, which leads to an enhancement of oxygen vacancies. The results of the MAH reaction show that the morphology and shape of CeO_2_ play an important role in the catalytic performance of the MAH reaction. The catalyst for the rod-like CeO_2_-R obtains a higher catalytic activity than the other two catalysts. It can be concluded that the higher catalytic performances of rod-like CeO_2_-R sample should be attributed to the higher dispersion of Ni particles, stronger support-metal interaction, more oxygen vacancies, and the lattice oxygen mobility. The research on the performances of morphology-dependent Ni/CeO_2_ catalysts as well as the relative reaction strategy of MAH will be remarkably advantageous for developing novel catalysts for MA hydrogenation.

## 1. Introduction

As high value-added solvents and intermediates, γ-butyrolactone (GBL), tetrahydrofuran (THF), and succinic anhydride (SA) are broadly employed in diverse industries including pesticides, machinery, military, plastics, and batteries. These fine chemicals can be prepared by means of catalytic hydrogenation of maleic anhydride (MA), which is a fundamental material that can be produced using butane in the petrochemical industry, using benzene from coal chemical’s primary product, and using 5-hydroxymethyl-furfaldehyde or furfaldehyde oxidation from biomass platform compounds [1,2,3,4]. In general, the procedure of maleic anhydride hydrogenation (MAH) comprises two sorts of hydrogenation procedures, namely the catalytic hydrogenation of C=C and C=O. These two hydrogenation procedures include similar reaction conditions (pressure and temperature) and can generate a mixture of various fine chemicals [5,6]. Therefore, the novel catalysts with remarkable catalytic activity and selectivity are highly desirable in order to reduce the contamination and further purification costs for synthesizing high-quality fine chemicals.

As a fundamental support material, ceria is capable of promoting the catalytic performances of catalysts by increasing the dispersion of the active metal particles and enhancing the support-metal interactions [7,8]. Furthermore, ceria is known as an oxygen buffer because of a fast and reversible transition between Ce^4+^ and Ce^3+^, which can provide various peculiar chemical and physical features. Hence, in a diverse range of catalytic reactions, including methane reforming, CO oxidation, NO catalytic reduction, and water-gas shift, ceria is extensively employed as an active catalyst or support material [9,10,11,12,13]. In our recent works [4,14], series Ni/CeO_2_ catalysts were synthesized and implemented for in-depth understanding of the important role of CeO_2_ in MA hydrogenation. By means of comparative research on support selection, the association of oxygen vacancies, Ni-CeO_2_ interaction, interface of M-CeO_2_ with the catalytic behaviors were thoroughly studied.

Very recently, a broad range of investigations has discovered that the feature of exposed lattice face has a notable impact on the catalytic behavior of CeO_2_. Moreover, morphology-dependent CeO_2_ has been deeply explored as a catalyst [15,16,17,18,19,20,21,22,23] and support [24,25,26,27,28,29,30,31,32,33] in various reactions, for instance in acetylene semi-hydrogenation, WGS reaction, and CO oxidation. Thus, the catalytic behaviors of CeO_2_-supported catalysts could be optimized and improved through modification of CeO_2_ morphology. Riley et al. [34] reported DFT calculation results on CeO_2_ (111) surface with oxygen vacancies and proposed the doping of ceria by Ni as a means of creating oxygen vacancies and enhancing the catalytic performance of the hydrogenation of acetylene. Vilé et al. [35] discovered that the cubic CeO_2_ was mostly exposed to (100) planes, while the particles of polyhedral CeO_2_ were mostly exposed to (111) planes, and the particles of polyhedral CeO_2_ demonstrated more considerable catalytic activity compared to the particles of cubic CeO_2_ in the C_2_H_2_ semi-hydrogenation reaction. Moreover, the exposed (110) plane of rod-like CeO_2_ illustrated more desirable catalytic performances in selective hydrogenation of nitroaromatic hydrocarbons [36]. Si and coworkers [24] prepared cubic, rod, and polyhedral shaped crystals of CeO_2_ with the aid of precipitation and deposition approaches. Subsequently, the preparation of Au/CeO_2_ catalysts was conducted by employing the synthesized CeO_2_ as support and implemented to WGS reaction. The following trend exhibits the catalytic activities of the three considered catalysts in the WGS reactions: rod-shaped Au/CeO_2_ > polyhedral Au/CeO_2_ ≫ cubic Au/CeO_2_. These investigations revealed that the exposed surface of CeO_2_ represented a substantial impact on the catalytic behaviors of the active metals since the dispersion of metals on the surfaces of the supports and the strong interaction of metal–support were regarded to be the key factors for the alterations in the physicochemical characteristics of the catalyst. As a result, the Ni/CeO_2_ catalyst possesses a remarkable catalytic activity in the hydrogenating process of maleic anhydride, and the morphology of CeO_2_ support also has an important effect on the chemical features and catalytic behavior of the catalyst, although researchers’ understanding of this aspect is limited.

In this study, based on the above discussions, three kinds of ceria, namely cube-like, rod-like, and particle-like, were prepared. By employing the impregnation technique, Ni species were dispersed on the surface of the particles of CeO_2_. The fundamental purpose of the current research is to study the potential effects of the CeO_2_ nano-crystal shape on the MAH reaction. It is revealed that the crystal plane and morphology of the CeO_2_-support show an apparent effect on the selectivity and conversion of MAH reaction. The catalyst supported over rod-like CeO_2_ obtains a substantially higher catalytic activity than the other two catalysts. The present research provides a deep understanding of nature and process of metal/oxide–carrier interactions and elucidates the optimization and synthesis technique of metal-supported catalysts; therefore, contributing to the preparation of MAH catalysts with higher product selectivity and catalytic activity.

## 2. Experimental Section

### 2.1. Catalysts Preparation

The utilized chemicals, namely, Ni(NO_3_)_2_*·*6H_2_O, Ce(NO_3_)_3_*·*6H_2_O, and NaOH were provided by the Sinopharm Chemical Reagent Co., Ltd. (Shanghai, China), and they were analytical grade and used as-purchased without pre-purification. The nanocrystals of CeO_2_ with various morphologies were prepared in accordance with Mai et al.’s procedure [17]. In general, the dissolution of 16.88 g NaOH was fulfilled in 30 mL water (6.0 mol/L), and the dissolution of 1.96 g Ce(NO_3_)_3_·6H_2_O was performed in 40 mL water (0.05 mol/mL). Subsequently, the solution of NaOH was increased dropwise into the solution of Ce(NO_3_)_3_ under stirring at ambient temperature. The prepared solution was sufficiently stirred for a further 30 min at ambient temperature and then transferred into a 100 mL Teflon bottle, which was tightly sealed and hydrothermally processed in a stainless-steel autoclave at 180 and 100 °C for 1 day accordingly for synthesizing CeO_2_ cubes (demonstrated as CeO_2_-C) and rods (demonstrated as CeO_2_-R). Following the cooling procedure, the resulting precipitate was collected, washed out with water, and dehydrated in a vacuum for 16 h at 80 °C. Afterwards, in a muffle oven, the obtained yellowish powder was calcined for 3 h at 500 °C for synthesizing the nanocrystals of CeO_2_-R and CeO_2_-C. In the case of CeO_2_ particles (denoted as CeO_2_-P), the synthesized process was practically the same as CeO_2_-R, except the added concentration of NaOH was about 0.01 mol/L. To prepare the Ni/CeO_2_ catalyst, the acquired CeO_2_ supports with three morphologies were moistened with corresponding volumes of the Ni(NO_3_)_2_·6H_2_O solution for achieving the theoretical Ni-loading of 5 wt%. The hydrated precursor was dehydrated during the night hours at 120 °C and then through calcination in air for 3 h at 450 °C for the production of NiO/CeO_2_ powder. Finally, the Ni (5 wt%)/CeO_2_ catalysts were achieved by means of reducing the NiO/CeO_2_ for 3 h at 350 °C, denoted as 5Ni/CeO_2_-R, 5Ni/CeO_2_-C, and 5Ni/CeO_2_-P, respectively.

### 2.2. Catalyst Characterizations

The outcomes of X-ray diffraction were provided through the powder diffraction of X-ray implementing a D8 Advance (Bruker, Billerica, MA, USA) (λ = 0.15418 nm, Cu Kα1 radiation) supplied with a Ni filter and Vantec detector. The findings were obtained within the 2*θ* zone of 10–80° through a scan speed at 3 °/min. The average crystallite size of the selected samples was evaluated by implementing the formula of Scherrer D = Kλ/Bcos*θ*, where K is 0.9. The materials morphologies were visualized by employing a Transmission Electron Microscope (TEM) (JEOL, JEM-2010, Tokyo, Japan). By taking advantage of the ultrasonic method, the specimens were dispersed in ethanol. The loading of Ni species of the catalysts was executed with the help of an inductively coupled plasma (ICP) spectroscopy instrument (iCAP 7400 ICP-OES, Thermo Fisher Scientific, Waltham, MA, USA). To characterize the property of a surface, the spectrum of Raman was obtained by using a Raman microscope with a 532 nm laser wavelength (HORIBA, Tokyo, Japan). The pore distribution and specific surface area of the catalysts, supports, and oxide precursors were determined through physisorption of N_2_ at −196 °C employing an ASAP-2020 device (Micromeritics, Norcross, GA, USA). The specimens were pre-degassed under vacuum at 250 °C prior to assessment. The specific surface area of the specimen was evaluated according to the Brunauer–Emmett–Teller (BET) method. The distribution of pore-size was assessed by considering the adsorption isotherms employing the Barett–Joyner–Halenda (BJH) approach.

The assessments of H_2_-TPR (hydrogen temperature programed reduction) and H_2_-TPD (Hydrogen temperature programmed desorption) were conducted on similar device employing the Micromeritics AutoChem II 2950 system supplemented with a thermal conductivity detector. X-ray photoelectron spectra (XPS) were recorded at ambient temperature employing a SCIENTIFIC ESCALAB 250 spectrometer supplemented with a standard Al-Kα (h = 1486.6 eV). By employing the carbon contamination (C1s, 284.6 eV), the calibration of the binding energies was accomplished. To fit XPS peaks, Lorentzian/Gaussian and Shirley functions were implemented together.

### 2.3. Catalytic Tests

The hydrogenation of MA at liquid was executed on the catalysts of Ni/CeO_2_ in a batch reactor at 5 MPa and 180 °C. Prior to MA hydrogenating, through the flow of pure H_2_ in a reactor at correspondent temperature, the catalysts were pre-reduced. The 4.9 g MA and 0.1 g reduced catalysts were mixed in a 100 mL autoclave comprising THF solvent; the mixed system was subsequently purged with N_2_ for the elimination of air. The reaction system was next heated to 180 °C with stirring at 500 rpm to diminish any possible mass transfer restriction. The pressure of hydrogen was kept at 5.0 MPa during the MAH reaction at the same time. The products of the reaction were scrutinized by employing a gas chromatograph (7890A, Agilent, Palo Alto, CA, USA). To verify precise separation of each component in the products, the programmed temperature was selected. The primary temperature of the oven was increased to 120 °C from 100 °C at a ramp of 5 °C min^−1^, and the temperatures of the detector and injector were 190 °C and 260 °C, respectively. The selectivity and conversion of MA to the product were evaluated considering the equations given below:$$X_{MA}\;\left(\%\right)=\frac{C_{GBL}+C_{SA}}{C_{GBL}+C_{SA}+C_{MA}}\times 100\%$$
$$S_{SA}\;\left(\%\right)=\frac{C_{SA}}{C_{SA}+C_{GBL}}\times 100\%$$
where *C_MA_*, *C_SA_*, and *C_GBL_* demonstrate the percent content of the reactant and products in the reaction sewage, respectively. *S_SA_* and *X_MA_* imply the *SA* selectivity and *MA* conversion.

## 3. Results and Discussion

### 3.1. Catalyst Characterization

Figure 1 demonstrates the images of HRTEM and TEM for the three nanomaterials studied. Figure 1A shows the images of TEM for the irregular nanoparticles of CeO_2_ with a less-uniform length within 10–20 nm. The image of HRTEM in Figure 1D shows the clear (111), (220), (200), and (311) lattice fringes with the interplanar spacing of 0.32, 0.28, 0.20, and 0.16 nm, respectively, meaning that the particles of CeO_2_-P are predominantly a hexahedral shape and surrounded by the (111) facet. Figure 1B exhibits the image of TEM for the CeO_2_ nanorods, with a less-uniform length within 20–150 nm and a uniform diameter in 10 ± 1.0 nm. Figure 1E illustrates the image of HRTEM for a CeO_2_ nanorod incorporated with a fast Fourier transform (FFT) assessment (inset). Considering the FTT assessment, two kinds of lattice fringe directions assigned to (200) and (220) are detected for the nanorods, which possess an interplanar spacing of 0.28 and 0.19 nm on the HRTEM image, respectively.

The nanorods demonstrate a 1D growth structure with a preferred growth direction along (220), and are surrounded by (200) planes, which are similar to the CeO_2_ nanorods synthesized under similar hydrothermal conditions by Mai and colleagues [17]. It is worth mentioning that the surface of CeO_2_ nanorods is rough, which implies the crystals have lower crystallinity and more defect sites on their surface. The image of TEM for the uniform nanocubes of CeO_2_ with the size of 10–50 nm is illustrated in Figure 1C. The image of HRTEM in Figure 1F incorporated with FFT assessment (inset) shows the apparent (220), (200), and (111) lattice fringes with the interplanar spacing of 0.19, 028, and 0.31 nm, respectively, indicating that the nanocubes of CeO_2_ are surrounded by the (200) planes.

The images of TEM in Figure 2 show that three various shaped nanomaterials of ceria maintain their intrinsic crystal shapes after the impregnation of Ni and further heat processing. According to the images of HRTEM, it could be observed that the species of Ni with exposed (111) planes are uniformly dispersed on the surface of CeO_2_ supports for all samples. The average sizes of Ni nanoparticles in 5Ni/CeO_2_-P, 5Ni/CeO_2_-R, and 5Ni/CeO_2_-C samples are 3.0, 2.0–3.0, and 5.0 nm, respectively. Therefore, most of the counted Ni in the samples are supported on the faces of the nanocrystals of CeO_2_ in comparison to their truncated corners and edges.

The XRD patterns for the as-prepared supports of CeO_2_ are shown in Figure 3A. The diffraction peaks related to Bragg for the CeO_2_ specimen presented at 28, 33, 47, and 56◦ can be described as (111), (200), (220), and (311) planes, which should belong to the fluorite-type structure of ceria with cubic crystalline (Fd3m, JCPDS file 34-0394) [37]. The weaker and wider peaks of diffraction for CeO_2_-Rs indicate a lower crystallinity and smaller crystallite size in comparison with the other samples. Owing to the impurity, no other diffraction peaks can be detected.

Following the introduction of nickel, the ceria remained in the primary face-centered cubic structure and no Ni diffraction peaks appeared (Figure 2B), which reveals that the Ni species anchored on CeO_2_ are smaller and highly dispersed, suggesting there is stronger interaction between CeO_2_ and Ni. Moreover, in comparison with pure ceria structures, the crystallite size of ceria (D) over Ni/CeO_2_ increases (Table 1), which is possibly correlated with the partial sintering during the procedure of thermal calcination. Based on the previous investigation [38], the microstrain (ε) of crystal is an assessment of lattice stress available in the materials due to lattice elongation, distortion, or contraction, which can be determined according to the broadening degree of XRD diffraction peak with pseudo-Voigt method. Thus, the number of inherent defects on the surface of three CeO_2_ supports can be qualitatively analyzed by comparing the value of microstrain. As shown in Table 1, the CeO_2_-R shows the highest value of microstrain both prior to and following the Ni loading, and the CeO_2_-C has the least value of microstrain. This order should be consistent with the reducibility (the oxygen vacancies concentration) for diverse CeO_2_ supports. Since there is a close association between the concentration of oxygen vacancies and the lattice strain, it is suggested that the asymmetrical five-coordinate structure of Ni/CeO_2_-R with the greatest strain could be unstable, which is desirable for the surface oxygen mobility. In contrast, the stability of the symmetrical eight-coordination structure of Ni/CeO_2_-C with the least strain could be considerable [38]. Due to the largest lattice strain and relatively highest concentration of oxygen defects, the 5Ni/CeO_2_-R should exhibit superior stability and catalytic activity compared with the other two samples for MA hydrogenation. Moreover, the specific surface areas achieved from the isotherms of N_2_ adsorption–desorption are also illustrated in Table 1. It can be detected that the 5Ni/CeO_2_-R and CeO_2_-R express the greatest S_BET_ values in comparison with the other samples. It should be noted that greater surface area of CeO_2_-R is in favor of the Ni dispersion on the support.

Raman spectroscopy is implemented for exploring the surface structure of the 5Ni/CeO_2_ catalysts and CeO_2_ supports. As shown in Figure 4A, the strong vibration mode (F2g, ~460 cm^−1^), because of the symmetrical stretching vibration of Ce-O bonds, predominates the Raman spectrum of CeO_2_. In addition to the F2g band, two wide bands at the regions of 1162 cm^−1^ and 590 cm^−1^ are also detected for the 5Ni/CeO_2_ and CeO_2_ specimens, which could be ascribed to the second order longitudinal optical mode (2LO), and the Frenkel defect-induced modes (D band), respectively [16]. The 2LO and D bonds are relevant to the existence of oxygen vacancies (Ovac) because of the existence of Ce^3+^ ions in the lattice of ceria, and the comparative ratio of intensity for I_D_/I_F2g_ reflects the oxygen vacancies concentration in ceria [39]. As shown in Figure 4, the values of I_D_/I_F2g_ and I_D_ + I_2LO_/I_F2g_ for Ni/CeO_2_ and CeO_2_ with different morphologies reduces in the following trend: CeO_2_-R > CeO_2_-P > CeO_2_-C. This represents that the amount of oxygen vacancies for the various ceria nanostructures differ as: nanorod > nanoparticle > nanocube.

Following loading the nickel (demonstrated in Figure 4B), the F2g modes red shifted from 460 cm^−1^ to 439 cm^−1^ with peak widening because of the strong interactions between CeO_2_ and Ni that resulted in the distortion of the lattice of CeO_2_ and generates electron-rich oxygen vacancies to maintain the system charge neutral [16]. In comparison with CeO_2_, all the samples of 5Ni/CeO_2_ show wider and stronger vibrations in the 2LO modes and D bands, and all of the ratios of intensity enhance sharply. Note that the incorporation of lower-valent cations such as Ni^2+^ in the CeO_2_ lattice yields defects as oxygen vacancies and dopant cations: the defect-induced (D) mode includes both the contributions due to oxygen vacancies and to cation substitution in the lattice (D1 and D2 bands, respectively). As shown in Figure 4B, D1and D2 components are clearly seen in the D band profile of Ni/CeO_2_-R and Ni/CeO_2_-P samples. The increase in (I_D_/I_F2g_) as well as the (I_D1_/I_D2_) intensity ratios, can be taken as an indication of a solid solution formation. This finding implies that the loading of Ni species facilitates the creation of oxygen vacancies (Ovac) thanks to metal substitution in the lattice of ceria, which is in a desirable consistency with the XRD achievements. Further, Ni/CeO_2_-R exhibits the largest value of I_D_/I_F2g_ among three samples, which subsequently reveals that the interactions between ceria rods, and nickel is stronger than other ceria structures. Thus, the morphology of CeO_2_ demonstrates an important influence on the synergistic interactions between ceria and nickel.

The H_2_-TPR files revealing the reducibility of catalysts are demonstrated in Figure 5. Four peaks of H_2_ consumption, namely α, β, γ, and δ are well-fitted through a Gauss-type function for three specimens of Ni/CeO_2_. The α and β peaks of H_2_ consumption at low temperature of 160 and 200 °C could be attributed to the reduction of the surface adsorbed oxygen species attached to the surface oxygen vacancies and considerably dispersed nano-crystallites of NiO for the catalysts, respectively. As explained before, the oxygen vacancy could be produced through the incorporation of Ni cations with the lattice of CeO_2_ and partial substitution of Ce^4+^ or/and Ce^3+^ cations to create solid solution. Moreover, Shan et al. and Li et al. successfully verified that the solid solution of Ni-Ce-O can be produced by means of the incorporation of the ions of Ni^2+^ into the lattice of CeO_2_, leading to oxygen vacancies and easily reduced oxygen species [40,41]. The oxygen vacancies can lead to charge imbalance and lattice distortion that is capable of adsorbing oxygen molecules on the surface of oxide. The adsorbed oxygen molecules are considerably reactive oxygen species and can be reduced easily by H_2_ at relatively lower temperatures in the range of 100 to 200 °C as shown in Figure 5.

The reduction peak γ could be attributed to the reduction of strongly interactive species of NiO with ceria support. It could be seen that the consumption of H_2_ for the γ peak on the Ni/CeO_2_-R is superior to that on the Ni/CeO_2_-C and Ni/CeO_2_-P catalysts, suggesting that a larger amount of strong interactive NiO is available on the surface of CeO_2_-R. The interaction of NiO-CeO_2_ obeys the following trend: Ni/CeO_2_-R > Ni/CeO_2_-P > Ni/CeO_2_-C. The reduction peak δ exhibited in the profiles of Ni/CeO_2_-C and Ni/CeO_2_-P is earmarked to the single step reduction of free NiO species, which accumulates on the surface and possesses weakly interaction with ceria support [42]. However, for the Ni/CeO_2_-R sample, the reduction behavior of the free NiO almost did not appear in the profile, which should be ascribed to the abundant defect sites on the CeO_2_ surface and high dispersion of NiO on the support in accordance with the XRD and Roman analysis. Hence, it can be concluded that the interaction between ceria rods and nickel is stronger in comparison with the other ceria structures. Consequently, the morphology of ceria support has a significant influence on the synergistic interactions between ceria and nickel.

XPS is a powerful tool for characterizing the surface composition of the catalysts and the valance state of the constituent elements, and also exploring the availability of oxygen vacancies. Figure 6A,B show the Ce3d and O1s core level XPS spectra of the reduced catalyst of Ni/CeO_2_. As shown in Figure 6A, the XPS spectrum for the Ce3d core level is deconvoluted into 10 Gaussian peaks and tagged according to the deconvolution conducted by Burroughs and colleagues [43]. The detected peaks labelled as U and V, U″ and V″ and U′′′ and V′′′ relate to 3d_3/2_ (3d_5/2_) and are characteristic of the final state of Ce^4+^3d, whilst U′ and V′ and U_0_ and V_0_ relate to 3d_3/2_ (3d_5/2_) for the final states of Ce^3+^ 3d [4,44]. Thus, the chemical valance of Ce on the surface of the reduced specimens is mostly an oxidation state of Ce^4+^, and a limited amount of Ce^3+^ co-existed.

Furthermore, it is obvious that there is an enhancement in the value of binding energy for the Ce3d_5/2_ component (883.9 eV) in comparison with the pure CeO_2_ (882.9 eV). This little shift could be described by the interactions between cerium oxide and nickel, meaning the incorporation of nickel into the cerium surface lattice [25]. Table 2 represents the comparative contributions of Ce^4+^ and Ce^3+^ evaluated by fitting the peaks and the areas under the fitted entities. The ratio of Ce^3+^/(Ce^4+^ + Ce^3+^) is observed to be dependent on the morphologies and the contents of Ce^3+^ in Ni/CeO_2_ decrease in the following trend: Ni/CeO_2_-R > Ni/CeO_2_-P > Ni/CeO_2_-C. In the previous reports [18,28], it has been reported that the existence of oxygen vacancies is capable of promoting the conversion of Ce^4+^ to Ce^3+^. Hence, considering both the XRD and Raman results, it can be defined through the creation of more surface oxygen vacancies over Ni/CeO_2_-R.

As shown in Figure 6B, the O1s XPS is fitted into three peaks and summarized in Table 2. The two lesser peaks of energy placed at 528.9 eV and 530.4 eV are assigned to lattice oxygen entities (O^2−^) binding to Ce^4+^ (O_II_) and Ce^3+^ (O_I_) [45], whilst the peak placed at 532.4 eV (O_III_) is attributed to the adsorbed oxygen entities (C–O species and water) on the surface of CeO_2_ [25]. The adsorption of CO_2_ and CO on the reduced state Ce^3+^ sites demonstrate a greater thermal stability compared to that on the sites of Ce^4+^ [46]. Consequently, the adsorbed oxygen is originated from carbonate entities trapped with the aid of oxygen vacancies. The oxygen vacancies content could be achieved from XPS comparative percentage of adsorbed oxygen. Table 2 represents the XPS findings of the lattice oxygen (O_II_ and O_I_) and the adsorbed oxygen (O_III_). According to Table 2, the ratio of O_I_/(O_I_ + O_II_ + O_III_) of these specimens illustrate the following trend: 5Ni/CeO_2_-R > 5Ni/CeO_2_-P > 5Ni/CeO_2_-C, which is consistent with the trend of the Ce^3+^ content. The great concentration of oxygen species on 5Ni/CeO_2_-R is because of its (110) plane with considerable chemical activity, which are active sites for the chemisorption of oxygen from H_2_O and CO_2_. Moreover, the ratio of O_III_/(O_I_ + O_II_ + O_III_) is able to determine this point. Figure 6C shows the Ni2p_3/2_ XPS spectra of reduced 5Ni/CeO_2_ catalysts. The Ni2p_3/2_ region is fitted into three peaks firstly by curve fitting using peaks and associated satellites in Figure 6C, and then three peaks are named with α, β, and γ, respectively. According to previous research, the peak α is assigned to Ni^0^, and peaks of β and γ refer to Ni^2+^. It can be seen that both Ni^0^ (α~852.7 eV) and Ni^2+^ (β~854.7 eV and γ~856.8 eV) coexist on the surface of Ni/CeO_2_ catalysts. Note that the peak area (α) of Ni^0^ for Ni/CeO_2_-R is larger than other two samples, which means more Ni^2+^ is reduced to Ni^0^ over the Ni/CeO_2_-R compared with other two samples, indicating that the Ni/CeO_2_-R has a higher reducibility of Ni species in this condition.

Hydrogen temperature programmed desorption (H_2_-TPD) is an advantageous approach for gaining deep insight in metal-support interaction and hydrogen activation on the catalysts. Figure 7 exhibits the profiles of H_2_-TPD for the 5Ni/CeO_2_ specimens with three configurations are deconvoluted into three peaks. The α peaks for the catalysts of Ni/CeO_2_ emerge in a remarkably similar temperature region (at 80 °C), which are attributed to the desorption of H_2_ up-taking surface oxygen vacancies in CeO_2_. The γ and β peaks placed in 100–200 °C are because of the desorption of H_2_ from the correspondent Ni entities. The β peaks could be assigned to ineffectively adsorbed dissociative H-entities taken up the interface of Ni-CeO_2_, which is close to oxygen vacancy on the support [4]. The γ peaks are correlated with the dissociated hydrogen entities attaching to the free particles of Ni. In the higher temperature region, the catalyst of Ni/CeO_2_ shows a single wide peak of H_2_-TPD emerging at 300 °C and tailing to further than 500 °C. This H_2_-TPD peak is because of the H_2_ adsorption in the subsurface layers of Ni or/and to the spillover of H_2_ [47].

In order to meticulously understand the difference in the capability of hydrogen activation between three various catalysts, quantitative assessment of the desorption of H was conducted and the results are given in Table 3. From Table 3, the amount of H_2_ desorption for the Ni/CeO_2_-R specimen is superior to other samples (such as interface-adsorbed H_β_, vacancy-adsorbed H_α_, and chemically adsorbed H_β_ on free Ni). The higher amount of H_2_ uptake for the Ni/CeO_2_-R explains it possesses higher dispersion and tinier size of Ni particle, which is in compliance with the results of XRD. Regarding the attribution of the peaks of H_2_-TPD, the dispersion of Ni could be quantified by the metal-related hydrogen activation (sum of γ- and β-H_2_), which is distinct from the total amount of H_2_ activation for the catalysts. As given in Table 3, the values of Ni dispersion for the 5Ni/CeO_2_ catalysts are shown to be diminishing according to the following order: 5Ni/CeO_2_-R > 5Ni/CeO_2_-P > 5Ni/CeO_2_-C. The results are in a favorable agreement with the TPR and Raman assessments. Furthermore, by means of comparative evaluations of H_2_-TPD and H_2_-TPR for three catalysts, we can deduce that the catalyst of Ni/CeO_2_-R represents more active sites for reversible adsorption/desorption of H-species, leading to fast conversion between H_2_ and the dissociated H species on the catalyst in MAH process and thus facilitating the MA conversion rate. Consequently, it can be estimated that the catalyst Ni/CeO_2_-R with rod shape has more activity in the reaction of hydrogenation.

### 3.2. Catalytic Performance

Figure 8A shows the conversion of MA (X_MA_) along the courses of hydrogenation on Ni/CeO_2_ catalysts in a batch reactor at 180 °C and the pressure of hydrogen was considered of about 5.0 MPa. As shown in Figure 8A, the 5Ni/CeO_2_-R catalyst is shown to present the highest activity for the conversion of MA, obtaining ~98% conversion of MA in 1 h. In the meantime, the conversion of MA on 5Ni/CeO_2_-C and 5Ni/CeO_2_-P in the same duration only obtains 90% and 92%, respectively. It is worth mentioning that the MA is completely converted into succinic anhydride (SA) on all three Ni/CeO_2_ catalysts (not shown) and exhibits 100% selectivity to SA within the 1.0 h continuous MAH without other products observed in the reaction, indicating it is inert for SA hydrogenolysis to other products in the current condition. The considerable difference of the MA conversion on three kinds of 5Ni/CeO_2_ catalysts can be attributed to the morphology and particle size, which can cause the difference of Ni dispersion and Ovac amount.

Figure 8B represents the −ln(1 − X_MA_) curves vs. time during the first 1 h, which are well-fitted conforming to the first-order kinetic law in respect of the conversion of MA on the metal-based catalysts [48]. The linear kinetic diagrams over three catalysts of Ni/CeO_2_ define that the hydrogenation of C=C obeys the quasi-first order reaction in respect of the conversion of MA. The MA hydrogenation rate coefficients (k) on these catalysts are determined according to the gradients of their linear plots and summarized in Figure 8B. The k values for 5Ni/CeO_2_-R, 5Ni/CeO_2_-P, and 5Ni/CeO_2_-C are around 0.0692, 0.0534, and 0.0415, respectively. The greater k value for 5Ni/CeO_2_-R specimen in comparison with others indicates that the 5Ni/CeO_2_-R is more reactive than other two catalysts.

Furthermore, the influence of the reaction temperature (393–453 K) on the activity of three catalysts in the hydrogenation of MA is carefully investigated. An Arrhenius diagram illustrating lnk vs. reaction temperature (1/T) is depicted in Figure 9. According to Figure 9, the constants of pseudo-first reaction rate lay entirely to the straight line, and an increase in activity with increasing temperature from 393 K to 453 K is observed. The apparent activation energies (Ea), evaluated from the slope of the Arrhenius plot straight line (shown in Figure 9), are revealed to be 47.93 ± 4.95 kJ/mol for 5Ni/CeO_2_-R, 50.69 ± 3.33 kJ/mol for 5Ni/CeO_2_-P and 58.22 ± 7.92 kJ/mol for 5Ni/CeO_2_-C, respectively. The values of Ea follow the order of: 5Ni/CeO_2_-R < 5Ni/CeO_2_-P < 5Ni/CeO_2_-C, which implies that the 5Ni/CeO_2_-R can represent the higher ability for MA hydrogenation.

From the above results, it can be seen that the surface structure, namely the morphology (crystal plane) of CeO_2_, is the key parameter influencing the interaction between Ni and CeO_2_ support. As shown in TEM images (Figure 1 and Figure 2), the CeO_2_-C, CeO_2_-R, and CeO_2_-P samples expose predominantly the (100), (110 or 220), and (111) planes, respectively. Because the typical CeO_2_ could be regarded as an array of cations creating a face-centered cubic lattice with oxygen ions situating at the tetrahedral interstitial sites, the Ni^2+^ is capable of penetrating easily into the lattices of (220) and (111) planes by placing in these sites, along with the capping oxygen atom to compensate the charge. This structural impact may lead to the differences in the synergistic interactions between CeO_2_ and Ni, thus affecting the catalytic behavior of Ni/CeO_2_ catalysts. The above analysis reasonably explained why Ni/CeO_2_-R shows a higher catalytic property compared to that of Ni/CeO_2_-P and Ni/CeO_2_-C.

Besides the interaction between CeO_2_ support and Ni, the oxygen vacancy (Ovac) is considered as another factor for enhancing the hydrogenation of C=C since the oxygen vacancy is capable of enriching the electron density of active metals, which promotes the electron donating capability and the dissociation of H2 [4,49]. In this study, the catalyst of Ni/CeO_2_-R possesses richer oxygen vacancies and hence can donate more electrons to the metallic nickel compared with the Ni/CeO_2_-C and Ni/CeO_2_-P, which shows higher activity of Ni/CeO_2_-R in the hydrogenation of MA. Thus, it can be concluded that the rod-shape of CeO_2_ can enhance the dispersion of metallic nickel, presenting stronger interaction between support and Ni and more oxygen vacancies, as a result the catalyst of 5Ni/CeO_2_-R shows higher reactivity in MAH compared with other two samples.

## 4. Conclusions

In this study, we have prepared three kinds of the catalysts of Ni/CeO_2_ with various morphologies of CeO_2_ support, namely cube-like, rod-like, and particle-like, and investigated the crystal-plane and morphology influences on catalytic behaviors in the reaction of maleic anhydride hydrogenation (MAH). The results of characterization demonstrate that nickel species can incorporate into the lattice of CeO_2_, and cause an increase in oxygen vacancies by occupying the empty sites. The catalytic features relate to the exposed plane and shape of the CeO_2_ supports. For all considered catalysts, the MA is able to be completely converted into succinic anhydride (SA), and represent 100% selectivity to SA in the current conditions. Among three catalysts, the Ni/CeO_2_-R exhibits excellent catalytic performance in stability and catalytic behavior, which is because of the stronger anchoring influence of CeO_2_ to nickel species. The oxygen vacancies concentration as well as the mobility of lattice oxygen within the Ni/CeO_2_ indicate the morphology dependencies. With the aid of reactivity assessments, the redox features and crystal structures of the catalysts achieved from various characterization methods, it can be deduced that the desirable behavior of 5Ni/CeO_2_-R is strongly associated with the higher dispersion of metallic nickel species, the higher availability of oxygen vacancies, and the stronger interaction between CeO_2_ and Ni. The obtained findings confirm that the catalytic performances and structures of Ni/CeO_2_ catalysts could be modulated through modifying the CeO_2_ support morphology. This investigation provides a deep understanding of the reaction MA hydrogenation by means of Ni/CeO_2_ catalysts.

## Figures and Tables

**Figure 1 nanomaterials-12-00762-f001:**
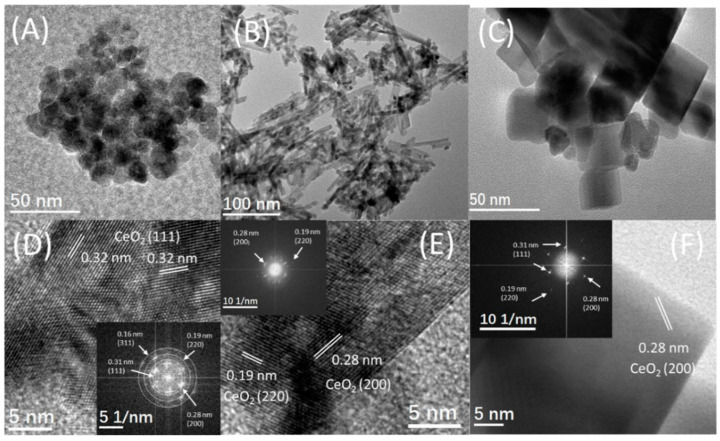
TEM, HRTEM, and FFT images of CeO_2_-P (**A**,**D**), CeO_2_-R (**B**,**E**), and CeO_2_-C (**C**,**F**) supports; inset is a fast Fourier transform (FFT) analysis.

**Figure 2 nanomaterials-12-00762-f002:**
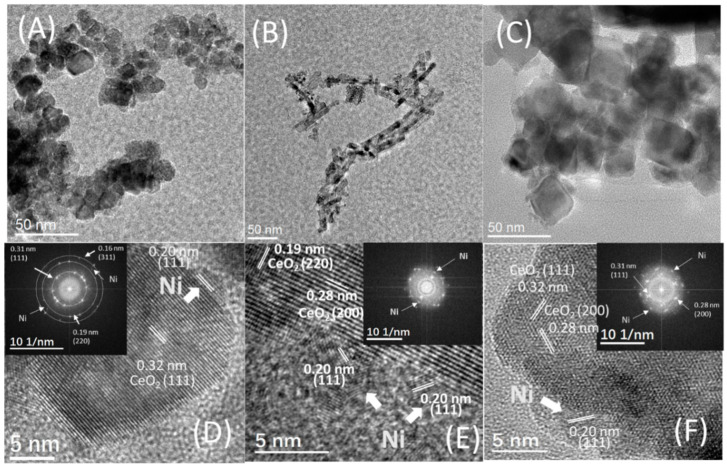
TEM, HRTEM, and FT images of 5Ni/CeO_2_-P (**A**,**D**), 5Ni/CeO_2_-R (**B**,**E**), and 5Ni/CeO_2_-C (**C**,**F**) supports; inset is a fast Fourier transform (FFT) analysis.

**Figure 3 nanomaterials-12-00762-f003:**
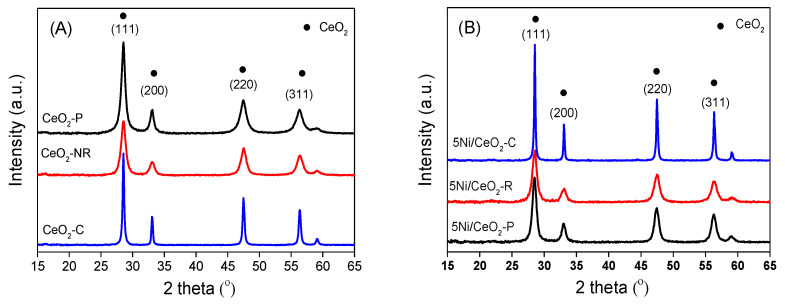
Patterns of XRD for different CeO_2_ supports (**A**) and 5Ni/CeO_2_ samples (**B**).

**Figure 4 nanomaterials-12-00762-f004:**
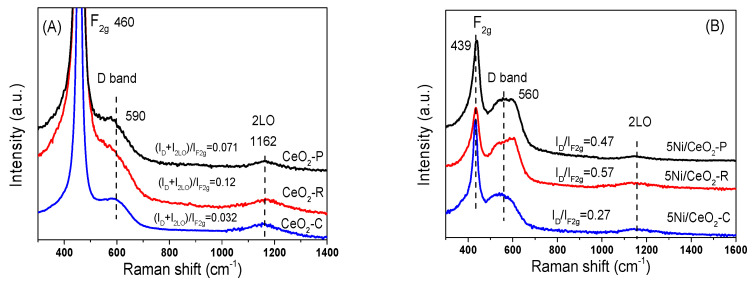
Raman spectra of different CeO_2_ support (**A**) and 5Ni/CeO_2_ catalysts (**B**).

**Figure 5 nanomaterials-12-00762-f005:**
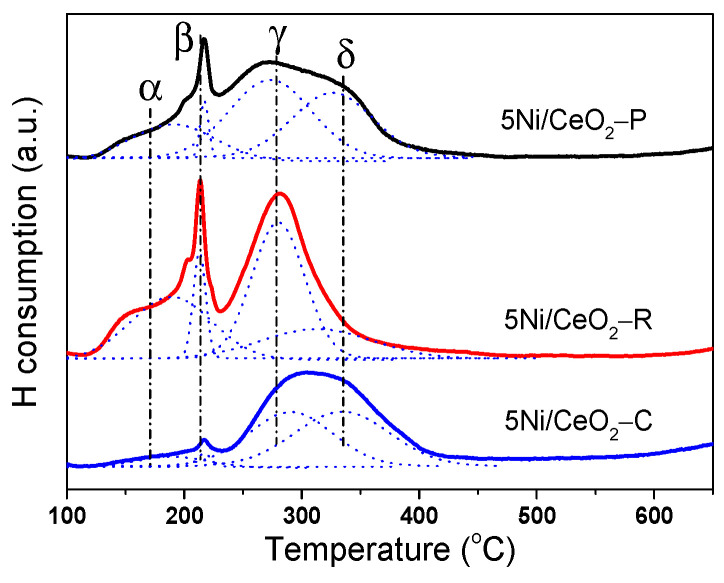
H_2_-TPR profiles of three 5Ni/CeO_2_ catalysts.

**Figure 6 nanomaterials-12-00762-f006:**
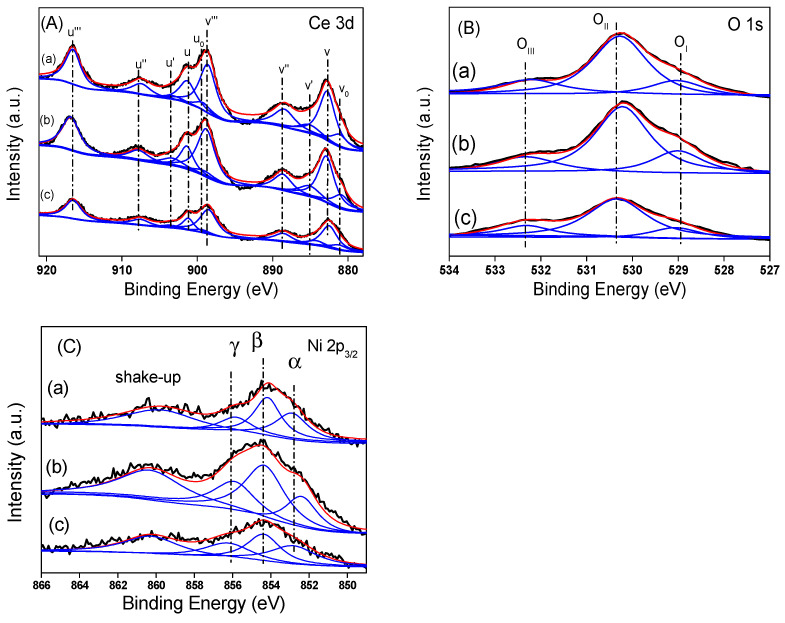
Spectra of XPS for (**A**) Ce 3d, (**B**) O 1s, and (**C**) Ni 2p_3/2_ for reduced 5Ni/CeO_2_. Catalysts, (**a**) 5Ni/CeO_2_-P, (**b**) 5Ni/CeO_2_-R, and (**c**) 5Ni/CeO_2_-C. The black line is primary curve, and the red and blue lines are fitted curves.

**Figure 7 nanomaterials-12-00762-f007:**
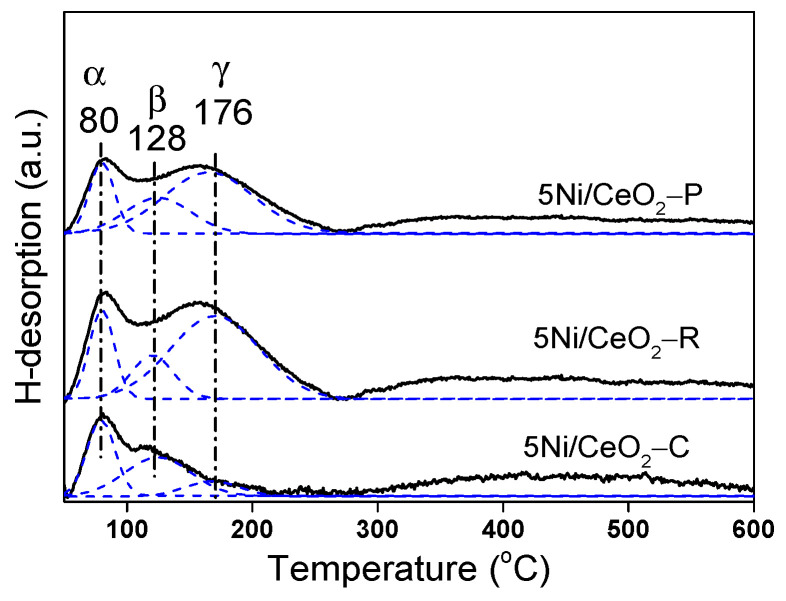
H_2_-TPD profiles of 5Ni/CeO_2_ catalysts.

**Figure 8 nanomaterials-12-00762-f008:**
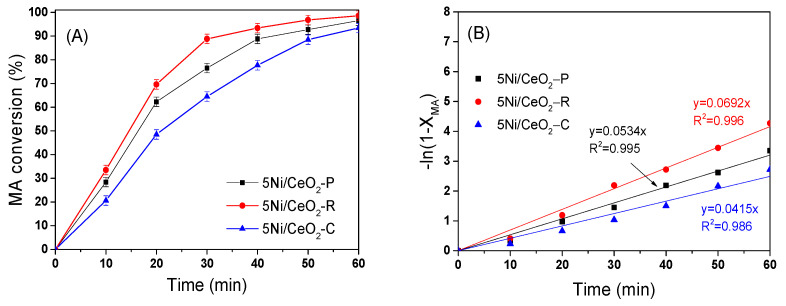
The conversion of maleic anhydride (MA) over 5Ni/CeO_2_ catalysts (**A**) and their −ln(1 − X_MA_) plots versus reaction time (**B**) at 180 °C and under 5 MPa of H_2_.

**Figure 9 nanomaterials-12-00762-f009:**
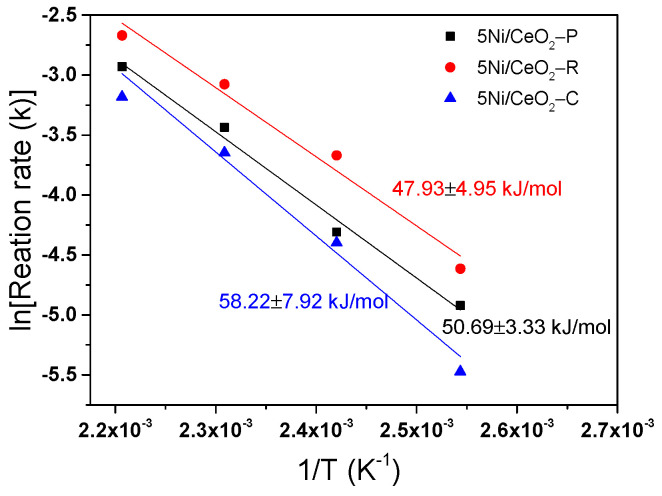
Arrhenius plot for the hydrogenation of maleic anhydride on three 5Ni/CeO_2_ catalysts and the relationship between ln(k_MA_) versus 1/T.

**Table 1 nanomaterials-12-00762-t001:** Physical parameters and structure of CeO_2_ supports and 5Ni/CeO_2_ catalysts.

Sample	Ni Loading (%)	S_BET_ (m^2^/g)	D (CeO_2_) (nm)	Microstrains (ε) (%)		
				(111)	(200)	(220)
CeO_2_-P	-	60.3	12.8	0.98	0.67	0.58
CeO_2_-R	-	80.2	10.6	1.04	0.79	0.62
CeO_2_-C	-	20.4	24.5	0.43	0.29	0.26
5Ni/CeO_2_-P	5.2	56.6	11.8	1.19	0.77	0.68
5Ni/CeO_2_-R	4.7	78.7	11.2	1.24	0.89	0.72
5Ni/CeO_2_-C	4.8	18.6	26.8	0.53	0.39	0.28

Note: The amount of metal loading was ascertained through ICP-OES. The specific surface area achieved from the isotherms of N_2_ adsorption–desorption. D(CeO_2_) represents the crystallite size of CeO_2_ phase, evaluated employing the equation of Scherrer to the (111) plane of ceria.

**Table 2 nanomaterials-12-00762-t002:** The quantitative XPS assessment of the 5Ni/CeO_2_ catalysts.

Sample	Ce^3+^/(Ce^4+^ + Ce^3+^) (%)	O_I_/(O_I_ + O_II_ + O_III_) (%)	O_III_/(O_I_ + O_II_ + O_III_) (%)
5Ni/CeO_2_-P	15.5	18.6	13.5
5Ni/CeO_2-_-R	18.7	20.5	25.4
5Ni/CeO_2_-C	14.6	12.3	9.4

**Table 3 nanomaterials-12-00762-t003:** H_2_ uptake and Ni dispersion on the 5Ni/CeO_2_ catalysts.

Sample	H_α_ (μmol/g)	H_β_ (μmol/g)	H_γ_ (μmol/g)	Ni Dispersion (%)
5Ni/CeO_2_-P	29.8	29.7	84.5	25.7
5Ni/CeO_2_-R	48.8	56.3	208.6	66.1
5Ni/CeO_2_-C	21.7	30.6	13.5	10.9

H_α_, H_β_, H_γ_ represent the amount of H_2_ desorbed at different temperatures. The Ni dispersion = (Ni_surf_/Ni_total_), Ni_total_ implies the total amount of Ni in the catalysts, and Ni_surf_ demonstrates the amount of surface-exposed Ni on the catalysts, considering H/Ni = 1 and Ni = 2 × (amount of H_2_ desorption).

## Data Availability

Not Applicable.

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
