# Peer review of "Crystal-Plane and Shape Influences of Nanoscale CeO2 on the Activity of Ni/CeO2 Catalysts for Maleic Anhydride Hydrogenation"

_nanomaterials, 2022, doi:10.3390/nano12050762_

Round 1
Reviewer 1 Report
The manuscript communicates results of a comprehensive experimental study aiming at uncovering dependencies of the activity of Ni/CeO2 catalysts on the arrangements of the nanostructured ceria support. Hydrogenation of maleic aldehyde (MA) is considered as an example reaction. A broad arsenal of state-of-the-art experimental techniques has been employed to characterize the three kinds of the prepared catalysts with Ni deposited on ceria nanomaterials presumably consisting of nanocubes, nanoparticles of other shapes and nanorods. The manuscript is written quite clearly and could be recommended for publication after the following issues are addressed:
- The role of oxygen vacancies in the catalytic hydrogenation of MA is discussed in several places of the manuscript and concluded to be similarly important with the roles of the strength with which Ni particles are anchored to ceria, with surface content of metallic Ni and Ce4+/Ce3+ interplay. In my opinion, the notable dependence of the catalytic activity on the concentration of oxygen vacancies in the support is not sufficiently documented by the provided experimental data. Thus, if no further experimental substantiation of the importance of oxygen vacancies can be provided, their role in the catalytic activity variation for the three catalysts under scrutiny has to be deemphasized in the discussion and conclusions of this work.
- Also, I was unable to find substantial experimental arguments supporting the statement (p. 12, lines 465-466): “The fundamental cause of the creation of interaction between CeO2 and Ni is the incorporation of Ni2+.” Please provide such arguments or rephrase/remove the sentence.
- It is unclear, how “the oxygen atoms of the lattice take part in the hydrogenation of MA.” (p. 12, lines 440-441)
- Incomprehensive/incomplete phrase (p. 2, line 58) “… calculations for the interaction of CeO2 (111) surface with oxygen vacancy …” Please amend.
- What are (p. 8, line 287) “simply reduced oxygen entities”?
- Several repeatedly used words of erroneous sense should be substituted by appropriate words. Among them: change “circumstances” to “conditions”, “achieved” to “obtained”, “meticulous (understanding)” to none of “detail (understanding)”, “accordingly” to “respectively”, “estimated” to “predicted” or “envisaged”, “obvious (activation energies)” to “apparent (activation energies)”, “thoroughly (converted)” to “completely (converted)”. As this list is incomplete, a careful proof-reading of the whole manuscript is highly recommended to apply the necessary corrections of the English wording.
Author Response
Dear editor and reviewer,
On behalf of all the authors, we really appreciate your kind evaluation and the reviewers’ review on our manuscript (ID: Nanomaterials-1556083). We have carefully studied the comments and suggestions. Each suggested revision and comment, brought forward by the reviewers was accurately incorporated and considered.
We have revised the manuscript carefully according to each comment. A detailed response, together with the corresponding changes made in our manuscript, is included in the following text (the changes in the revised manuscript are highlighted in red). The whole revision has also been experienced a careful proof-reading by a native English-speaking expert to ensure its English is good enough for publication.
We really appreciate your time and effort given to the manuscript.
Best regards,
Dr. Prof. Hao Wang
Engineering Research Center of Ministry of Education for Fine Chemicals
School of Chemistry and Chemical Engineering
Shanxi University
Response to Reviewer
The manuscript communicates results of a comprehensive experimental study aiming at uncovering dependencies of the activity of Ni/CeO2 catalysts on the arrangements of the nanostructured ceria support. Hydrogenation of maleic aldehyde (MA) is considered as an example reaction. A broad arsenal of state-of-the-art experimental techniques has been employed to characterize the three kinds of the prepared catalysts with Ni deposited on ceria nanomaterials presumably consisting of nanocubes, nanoparticles of other shapes and nanorods. The manuscript is written quite clearly and could be recommended for publication after the following issues are addressed:
Comment 1: The role of oxygen vacancies in the catalytic hydrogenation of MA is discussed in several places of the manuscript and concluded to be similarly important with the roles of the strength with which Ni particles are anchored to ceria, with surface content of metallic Ni and Ce4+/Ce3+ interplay. In my opinion, the notable dependence of the catalytic activity on the concentration of oxygen vacancies in the support is not sufficiently documented by the provided experimental data. Thus, if no further experimental substantiation of the importance of oxygen vacancies can be provided, their role in the catalytic activity variation for the three catalysts under scrutiny has to be deemphasized in the discussion and conclusions of this work.
Response: Thanks so much for the comment of reviewer. As pointed out by reviewer, the main factor affecting the catalytic performance of catalyst is the active metal Ni in the process of MA hydrogenation, including the loading amount of Ni, the dispersion of Ni particles, and the interaction between the support and metal Ni and so on. Oxygen vacancy and other factors are secondary, and only play a supplementary role. In this paper, the aim is to investigate the morphology influences of CeO2 support on the activity of Ni/CeO2 catalysts for MA Hydrogenation, and we try to explain the key reasons leading to the difference in catalytic performance. As discussed in the manuscript, oxygen vacancy should play an important role for the difference of catalytic performance of the three catalysts. This is because the different morphologies of CeO2 will lead to the change of surface defect sites and oxygen vacancies, and then cause the change of Ni dispersion on the surface of CeO2. Therefore, here the role of oxygen vacancy is emphasized mainly to explain the reason for the variation of catalytic activity caused by three kinds of support with different morphologies. For the whole MA hydrogenation reaction, there is no doubt that the decisive factor affecting the catalytic activity of Ni/CeO2 catalysts is the active metal Ni.
Comment 2: Also, I was unable to find substantial experimental arguments supporting the statement (p. 12, lines 465-466): “The fundamental cause of the creation of interaction between CeO2 and Ni is the incorporation of Ni2+.” Please provide such arguments or rephrase/remove the sentence.
Response: Thanks for the comment. The relative description in the manuscript is not precise here. According to the suggestion of reviewer, this sentence was removed in the revised manuscript to avoid the possible confusion.
Comment 3: It is unclear, how “the oxygen atoms of the lattice take part in the hydrogenation of MA.” (p. 12, lines 440-441)
Response: Thanks for the comment. Here the relative expression about oxygen atom is inappropriate and unclear. In order to avoid possible misunderstanding and confusion, the relative sentences were deleted in the revision.
Comment 4: Incomprehensive/incomplete phrase (p. 2, line 58) “… calculations for the interaction of CeO2 (111) surface with oxygen vacancy …” Please amend.
Response: Thanks for the suggestion of reviewer. According to the suggestion, this sentence was amended carefully in the revised manuscript (see Page 2, line 58-60 in the revision and also shown below).
“Riley et al. [34] reported a DFT calculation results on CeO2 (111) surface with oxygen vacancies and proposed the doping of ceria by Ni as a means to create oxygen vacancies and enhance the catalytic performance in the hydrogenation of acetylene”.
Comment 5: What are (p. 8, line 287) “simply reduced oxygen entities”?
Response: Sorry for the improper expression. According to the reviewer’s suggestion, the relative phrase has been examined and correlated to “easily reduced oxygen species” in the revised manuscript (see Page 7, line 282 and also shown below).
“Li et al. and Shan et al. also confirm that the Ni-Ce-O solid solution could be generated by the incorporation of Ni2+ ions into CeO2 lattice, resulting in oxygen vacancies and easily reduced oxygen species”.
Comment 6: Several repeatedly used words of erroneous sense should be substituted by appropriate words. Among them: change “circumstances” to “conditions”, “achieved” to “obtained”, “meticulous (understanding)” to none of “detail (understanding)”, “accordingly” to “respectively”, “estimated” to “predicted” or “envisaged”, “obvious (activation energies)” to “apparent (activation energies)”, “thoroughly (converted)” to “completely (converted)”. As this list is incomplete, a careful proof-reading of the whole manuscript is highly recommended to apply the necessary corrections of the English wording.
Response: We really appreciate the useful suggestions of reviewer. According to the suggestion, the improper words mentioned by reviewer have been substituted by the appropriate words, and the whole manuscript has also been experienced a careful proof-reading by a native English-speaking expert to ensure its English is good enough for publication.

Reviewer 2 Report
In the manuscript entitled "Crystal-Plane and Shape Influences of Nanoscale CeO2 on the 3 Activity of Ni/CeO2 Catalysts for Maleic Anhydride Hydro-4 genation" Yongxiang Zhao al. reported the study of three sorts of ceria, namely CuO2 be-like, rod-like, and particle-like were prepared and by employing the impregnation technique, Ni species were dispersed on the surface of the particles of CeO2. The topic is interesting, and the manuscript was prepared carefully, at least from the formal point of view. Language is acceptable. However, in several places, too wordy and, on a few occasions, the meaning is unclear. They should improve the English of the manuscript by proofreading. I recommend the publication of the work.
Author Response
Dear editor and reviewer,
On behalf of all the authors, we really appreciate your kind evaluation and the reviewers’ review on our manuscript (ID: Nanomaterials-1556083). We have carefully studied the comments and suggestions. Each suggested revision and comment, brought forward by the reviewers was accurately incorporated and considered.
We have revised the manuscript carefully according to each comment. A detailed response, together with the corresponding changes made in our manuscript, is included in the following text (the changes in the revised manuscript are highlighted in red). The whole revision has also been experienced a careful proof-reading by a native English-speaking expert to ensure its English is good enough for publication.
We really appreciate your time and effort given to the manuscript.
Best regards,
Dr. Prof. Hao Wang
Engineering Research Center of Ministry of Education for Fine Chemicals
School of Chemistry and Chemical Engineering
Shanxi University
Response to Reviewer 2
In the manuscript entitled "Crystal-Plane and Shape Influences of Nanoscale CeO2 on the Activity of Ni/CeO2 Catalysts for Maleic Anhydride Hydrogenation" Yongxiang Zhao al. reported the study of three sorts of ceria, namely CuO2 be-like, rod-like, and particle-like were prepared and by employing the impregnation technique, Ni species were dispersed on the surface of the particles of CeO2. The topic is interesting, and the manuscript was prepared carefully, at least from the formal point of view. Language is acceptable. However, in several places, too wordy and, on a few occasions, the meaning is unclear. They should improve the English of the manuscript by proofreading. I recommend the publication of the work.
Response: We are grateful for the time and effort of the reviewer on our manuscript. According to the suggestion, the improper words and wordy sentences have been substituted and corrected in the revision, and the whole manuscript has also been experienced a careful proof-reading by a native English-speaking expert to ensure the language is proper for publication. The corresponding changes made in the manuscript are highlighted in red in the revision.

Reviewer 3 Report
In the paper “Crystal-plane and shape influences of nanoscale CeO2 on the activity of Ni/CeO2 Catalysts for Maleic Anhydride Hydrogenation” the Authors focused on the effect of CeO2 morphology on the catalytic performances of Ni/CeO2 catalysts for Maleic Anhydride Hydrogenation.
The role of CeO2 morphology on the stability and reactivity of surface oxygen species and on defects formation in ceria and doped ceria materials is widely recognized in the literature and properly reviewed in the Introduction by the Authors.
Even though the topic is of great interest and the experimental design appropriate to test the system catalytic performances, the manuscript lack of novelty, results are presented and discussed with convoluted sentences often using improper terms that do not allow the reader to easily follow the
text and the originality of the research results.
Some comments
Catalysts characterization
The experimental design and methods should be described using proper scientific terminology:
- …“The average crystallite scales” of the selected entities → “average crystallite size….”
- …Brunauer–Emmett–Teller (BET) “process” → “method”
- …“The spectra of XPS” … →X-ray photoelectron (XPS) spectra were recorded
-… By employing “the containment carbon” (C1s, 284.6 eV) → “carbon contamination”
- …the calibration of the “energies of binding” → “binding energies”…
Results
CeO2 morphology
- …Worth mentioning that the surface of CeO2 nanorods is less smooth, which implies “the lower crystallinity of crystals” … → explain properly “lower crystallinity of crystals”
- …Furthermore, according to the images of HRTEM demonstrated in Fig. 1, it can observed that the nature of as-obtained CeO2 nanorods, nanocubes, and nanoparticles is “single-crystalline”…. →explain properly “single-crystalline”
- ….The patterns of XRD for …. The “peaks of diffraction related to Bragg” for the CeO2 specimen presented at 56, 47, 33, 210 and 28â—¦ can be described as (311), (220), (200), and (111) planes…, → XRD patterns….”diffraction peaks” …at…. → peak positions must follow the order shown in the diffraction patterns in Fig. 3: increasing 2 theta values
- According to Table 1, the CeO2-R illustrates the “greatest microstrain” both prior to and following the Ni loading, and the CeO2-C demonstrated the “least strain” → explain properly
Raman spectroscopy
- … Fig. 4A, the “potent vibration mode” → “strong vibration mode”
- … In addition to the F2g band, two “widening bands” → “wide bands”
- …The 2LO and D “bond” → bands
- … the oxygen vacancy could be produced through the incorporation of Ni entities with the lattice of CeO2 and partial substitution of Ce4+ or/and Ce3+ cations to create solid solution…. → Note that the incorporation of lower-valent cations such as Ni2+ in the CeO2 lattice yields defects as oxygen vacancies and dopant cations: the defect-induced (D) mode includes both the contributions due to oxygen vacancies and to cation substitution in the lattice (D1 and D2 bands, respectively). As shown in Fig. 4B, D1and D2 components are clearly seen in the D band profile of Ni/CeO2-R and Ni/CeO2-P samples. The increase of (ID/IF2g) as well as the (ID1/ID2) intensity ratios, can be taken as an indication of a solid solution formation.
XPS
-...Fig. 6(A) and (B) illustrate the “ Ce O1s and 3d core level XPS” → Ce 3d and O1s core level
XPS
-...The oxygen entities available in the Ni/CeO2 catalyst are ascertained through the evaluation of O1s XPS. According to Fig. 6(B), the O1s XPS is well-fitted into three peaks…
→ the discussion should be more concise, avoiding too much speculation
-…The Ni2p3/2 XPS spectra for the reduced catalysts of 5Ni/CeO2 is represented in Fig. 6C. Based on this figure, both Ni 2+ (β~854.7 eV and γ~856.8 eV) and Ni0 (α~852.7 eV) coexist on the surface of Ni/CeO2 catalysts…
→ The Ni2p3/2 region should be reproduced by curve fitting using peaks and associated satellites, rather than using peaks and one satellite (Fig. 6C), according to spectroscopic rules
→ The Ni2p3/2 α nd β species should be assigned.
Discussion
This paragraph is useless as it represents a repetition of what was said in the result section.
Author Response
Dear editor and reviewer,
On behalf of all the authors, we really appreciate your kind evaluation and the reviewers’ review on our manuscript (ID: Nanomaterials-1556083). We have carefully studied the comments and suggestions. Each suggested revision and comment, brought forward by the reviewers was accurately incorporated and considered.
We have revised the manuscript carefully according to each comment. A detailed response, together with the corresponding changes made in our manuscript, is included in the following text (the changes in the revised manuscript are highlighted in red). The whole revision has also been experienced a careful proof-reading by a native English-speaking expert to ensure its English is good enough for publication.
We really appreciate your time and effort given to the manuscript.
Best regards,
Dr. Prof. Hao Wang
Engineering Research Center of Ministry of Education for Fine Chemicals
School of Chemistry and Chemical Engineering
Shanxi University
Response to Reviewer
In the paper “Crystal-plane and shape influences of nanoscale CeO2 on the activity of Ni/CeO2 Catalysts for Maleic Anhydride Hydrogenation” the Authors focused on the effect of CeO2 morphology on the catalytic performances of Ni/CeO2 catalysts for Maleic Anhydride Hydrogenation. The role of CeO2 morphology on the stability and reactivity of surface oxygen species and on defects formation in ceria and doped ceria materials is widely recognized in the literature and properly reviewed in the Introduction by the Authors.
Even though the topic is of great interest and the experimental design appropriate to test the system catalytic performances, the manuscript lack of novelty, results are presented and discussed with convoluted sentences often using improper terms that do not allow the reader to easily follow the text and the originality of the research results.
Response: Thanks a lot for the reviewer’s comments. The comments are very helpful for further strengthening our manuscript. As for the novel aspect of this research, one important point should be highlighted. Though many studies have focused on the investigation of CeO2 morphology and also shown that the crystal shape of CeO2 support plays an important role in some chemical reactions. Until now, there is very limited knowledge for the effect of CeO2 morphology on the chemical state and catalytic performance for maleic anhydride hydrogenation (MAH). In this paper, three different morphologies of ceria (rod, cube, and particle) were prepared. We deeply investigated the potential effect of the CeO2 shape on the MAH reaction and explained the essential reason about the difference of catalytic performance for three catalysts. The acquired results not only have provided novel insights into the metal/oxide-support interaction, but also have demonstrated morphology engineering as an effective strategy to optimize the structure and catalytic performance of supported catalysts.
According to the reviewer’s comment, we have checked the manuscript and refined the language carefully. The convoluted sentences and improper terms in the manuscript have also been corrected thoroughly. A detailed response, together with the corresponding changes made in our manuscript, is included in the following text (the changes in the revised manuscript are highlighted in red).
Some comments
Catalysts characterization
The experimental design and methods should be described using proper scientific terminology:
- …“The average crystallite scales” of the selected entities → “average crystallite size….”
- …Brunauer–Emmett–Teller (BET) “process” → “method”
- …“The spectra of XPS” … →X-ray photoelectron (XPS) spectra were recorded
-… By employing “the containment carbon” (C1s, 284.6 eV) → “carbon contamination”
- …the calibration of the “energies of binding” → “binding energies”…
Response: Many thanks for the reviewer’s comments.
- “The average crystallite scales” has been corrected to “the average crystallite size” in the revision.
- Brunauer–Emmett–Teller (BET) “process” has been corrected to “method” in the revision.
- The phrase“The spectra of XPS” has been corrected to “X-ray photoelectron spectra (XPS) were recorded” in the revision.
- The phrase “containment carbon” (C1s, 284.6 eV) has been modified to “carbon contamination” in the revision.
- The phrase “energies of binding” has been revised to “binding energies” in the revision.
Results
CeO2 morphology
- …Worth mentioning that the surface of CeO2 nanorods is less smooth, which implies “the lower crystallinity of crystals” … → explain properly “lower crystallinity of crystals”
Response: It is known that the crystal surface should be smooth and perfect if the growth process of crystal is completed very well and has a higher crystallinity. Here, compared with other two samples, the surface of CeO2 nanorods is less smooth and rough, which means the crystal growth of rod-like sample is incomplete and the crystallinity of crystal is lower. In order to avoid the possible confusion, we have been corrected the sentence to “which implies the crystals have the lower crystallinity and more defect sites on the surface” in the revision.
- …Furthermore, according to the images of HRTEM demonstrated in Fig. 1, it can observed that the nature of as-obtained CeO2 nanorods, nanocubes, and nanoparticles is “single-crystalline”…. →explain properly “single-crystalline”
Response: Thanks for the comment of reviewer. Here, the statement of single-crystalline should be inaccurate and unprecise, since it is very hard to judge the single-crystalline of crystal only by TEM images. We have deleted the relative sentence in the revision to avoid possible confusion.
- ….The patterns of XRD for …. The “peaks of diffraction related to Bragg” for the CeO2 specimen presented at 56, 47, 33 and 28â—¦ can be described as (311), (220), (200), and (111) planes…, → XRD patterns….”diffraction peaks” …at…. → peak positions must follow the order shown in the diffraction patterns in Fig. 3: increasing 2 theta values.
Response: Thanks a lot for the comments. According to the comments, the phrases “The patterns of XRD” and “peaks of diffraction”have been corrected to “XRD patterns” and “diffraction peaks” in the revision. The statement for peak position also corrected to “The diffraction peaks related to Bragg for the CeO2 specimen presented at 28, 33, 47, and 56â—¦ can be described as (111), (200), (220), and (311) planes” in the revision (shown in page 5).
- According to Table 1, the CeO2-R illustrates the “greatest microstrain” both prior to and following the Ni loading, and the CeO2-C demonstrated the “least strain” → explain properly
Response: Based on the previous investigation [38], the microstrain (ε) of crystal is an assessment of lattice stress available in the materials due to lattice elongation, distortion, or contraction, which can be determined by the broadening degree of XRD diffraction peak with pseudo-Voigt method. So the number of inherent defects on the surface of three CeO2 supports can be qualitatively analyzed by comparing the value of microstrain. As shown in Table 1, the CeO2-R shows the highest value of microstrain both prior to and following the Ni loading, and the CeO2-C has the least value of microstrain. The relative explanation has added to the revision in the page 6.
Raman spectroscopy
- … Fig. 4A, the “potent vibration mode” → “strong vibration mode”
- … In addition to the F2g band, two “widening bands” → “wide bands”
- …The 2LO and D “bond” → bands
Response: Thank a lot for the reviewer’s comment and very sorry for the improper terms and words. According to the reviewer’s comments, the above errors have been corrected in the revision, such as the “potent vibration mode” corrected to “strong vibration mode”, the“widening bands” corrected to “wide bands”, and the“bond” to “bands”, respectively.
- … the oxygen vacancy could be produced through the incorporation of Ni entities with the lattice of CeO2 and partial substitution of Ce4+ or/and Ce3+ cations to create solid solution…. → Note that the incorporation of lower-valent cations such as Ni2+ in the CeO2 lattice yields defects as oxygen vacancies and dopant cations: the defect-induced (D) mode includes both the contributions due to oxygen vacancies and to cation substitution in the lattice (D1 and D2 bands, respectively). As shown in Fig. 4B, D1and D2 components are clearly seen in the D band profile of Ni/CeO2-R and Ni/CeO2-P samples. The increase of (ID/IF2g) as well as the (ID1/ID2) intensity ratios, can be taken as an indication of a solid solution formation.
Response: We really appreciate the comment of the reviewer on the result of Raman spectra. The comments are very helpful for further understanding and strengthening the defect-induced (D) mode. We have added these sentences to the revised manuscript in page 7 according to the comments.
XPS
-...Fig. 6(A) and (B) illustrate the “Ce O1s and 3d core level XPS” → Ce 3d and O1s core level XPS
Response: Sorry for this error. According to the comment, the phrase “Ce O1s and 3d core level XPS” has been corrected to “Ce3d and O1s core level XPS” in the revision.
-...The oxygen entities available in the Ni/CeO2 catalyst are ascertained through the evaluation of O1s XPS. According to Fig. 6(B), the O1s XPS is well-fitted into three peaks…→ the discussion should be more concise, avoiding too much speculation.
Response: Thanks for the comment. We have modified the sentence to make the revision more concise and avoid possible speculation according to the comment.
-…The Ni2p3/2 XPS spectra for the reduced catalysts of 5Ni/CeO2 is represented in Fig. 6C. Based on this figure, both Ni2+ (β~854.7 eV and γ~856.8 eV) and Ni0 (α~852.7 eV) coexist on the surface of Ni/CeO2 catalysts…
→ The Ni2p3/2 region should be reproduced by curve fitting using peaks and associated satellites, rather than using peaks and one satellite (Fig. 6C), according to spectroscopic rules
Response: Thanks a lot for the comment. As mentioned by reviewer, we exactly determined the three peaks of Ni2p3/2 region by curve fitting using peaks and associated satellites firstly, and then named the three peaks with α, β and γ. Sorry for the improper description here. We have revised the relative sentence in the revision (see Page 9 and also shown below).
“Fig. 6C shows the Ni2p3/2 XPS spectra of reduced 5Ni/CeO2 catalysts. The Ni2p3/2 region is fitted into three peaks firstly by curve fitting using peaks and associated satellites in Fig. 6C, and then three peaks are named with α, β and γ, respectively. According to the previous research, the peak α is assigned to Ni0, and peaks of β and γ refer to Ni2+. It can be seen that both Ni0 (α~852.7 eV) and Ni2+ (β~854.7 eV and γ~856.8 eV) coexist on the surface of Ni/CeO2 catalysts. Note that the peak area (α) of Ni0 for Ni/CeO2-R is larger than other two samples, which means more Ni2+ is reduced to Ni0 over the Ni/CeO2-R compared with other two samples, indicating that the Ni/CeO2-R has a higher reducibility of Ni species in this condition.”
→ The Ni2p3/2 α and β species should be assigned.
Response: According to the comment of reviewer, the peak α is assigned to Ni0 species, and peaks of β and γ refer to Ni2+ species. The relative sentence has been added to the revision.
Discussion
This paragraph is useless as it represents a repetition of what was said in the result section.
Response: The purpose of this paragraph is to further explain and discuss the factors that affect catalytic performance of catalyst. As pointed out by reviewer, there is much repetition between this paragraph and the previous chapter. According to the suggestion of reviewer, we have simplified and rewrote this paragraph, and incorporated this part into the above sections in the revision.
